# OvisOCR: End-to-End Document Parsing via Aligning Specialized Perception with General Reasoning

**Jun-Peng Jiang** [1 2 3 *]  **Shiyin Lu** [3]  **An-Yang Ji** [1 2 3 *]  **Yinglun Li** [3]
**Qing-Guo Chen** [3]  **Zhao Xu** [3]  **Weihua Luo** [3]  **Kaifu Zhang** [3]  **De-Chuan Zhan** [1 2]  **Han-Jia Ye** [1 2]

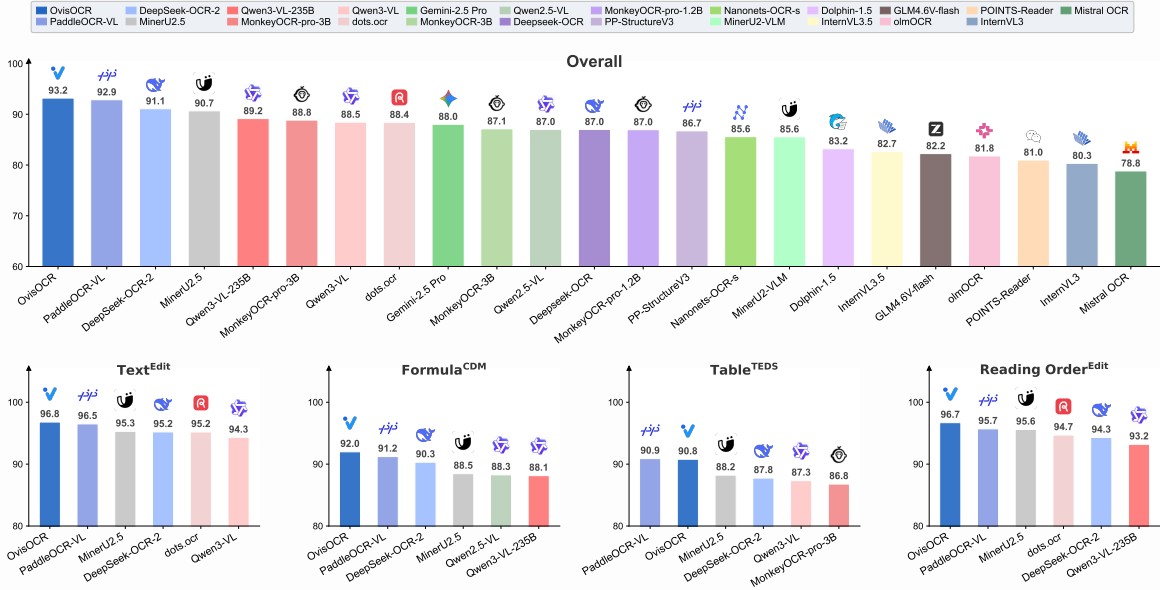

*Figure 1.* Performance comparison of OvisOCR and other models in OmniDocBench v1.5 (Ouyang et al., 2025).

## Abstract

This paper presents OvisOCR, a lightweight and strictly end-to-end Multimodal Large Language Model (MLLM) tailored for document parsing. Unlike current methods that rely on complex "Crop-OCR-Merge" cascades to handle high-resolution inputs, OvisOCR directly maps full-page visual signals to structured Markdown without localized slicing or layout detection dependencies. Through extensive evaluations on the OmniDocBench, OvisOCR achieves top-tier performance among all compared methods in all aspects, demonstrating that a compact E2E model

can effectively "digest" the capabilities of intricate pipelines and surpass specialized and general methods. Technically, OvisOCR unifies fine-grained recognition and semantic correction in a single model by leveraging supervision bootstrapped from strong OCR engines and refined via general-model-based data cleaning. To balance the performance across diverse document constituents, we design category-specific reward mechanisms for distinct element types, such as dense text, complex tables, and formulas, and ensure the model enhances its formatting strengths for each modality concurrently. This approach effectively resolves the optimization conflict, guaranteeing that improvements in structural layout parsing do not come at the expense of omitting fine-grained textual details. Empirical results confirm that OvisOCR eliminates the error propagation inherent in split-and-merge architectures, offering a streamlined path for future document intelligence. Our model is available at https://huggingface.co/ATH-MaaS/OvisOCR.

*Work done during an internship at Alibaba Group [1]School of Artificial Intelligence, Nanjing University, China [2]National Key Laboratory for Novel Software Technology, Nanjing University, China [3]Alibaba Group. Correspondence to: Han-Jia Ye <yehj@lamda.nju.edu.cn>.

*Proceedings of the 43rd International Conference on Machine Learning*, Seoul, South Korea. PMLR 306, 2026. Copyright 2026 by the author(s).

# 1. Introduction

Documents (e.g., forms, reports, invoices, academic papers) contain rich and heterogeneous information, spanning natural language text, tables, mathematical formulas, and various layout structures. Document parsing (Kim et al., 2022; Hwang et al., 2021; Zhang et al., 2024; Ding et al., 2024) aims to faithfully extract and serialize all such contents from a document image into a structured representation (e.g., Markdown/JSON), following the human reading order, producing an output that preserves both fine-grained textual fidelity and the global document structure.

Document parsing has been studied for years. Traditional systems (Wang et al., 2024; Vik, 2025; Cui et al., 2025b) typically follow a modular pipeline, combining OCR with layout analysis, table/formula extraction, and reading-order recovery, and then aggregating the results into structured formats for downstream use. Recently, multimodal large language models (MLLMs) (Chen et al., 2024; Bai et al., 2025; Lu et al., 2025) have begun to reshape this field by reframing document parsing as a single generative task: conditioned on the document image, the model directly generates structured outputs such as Markdown/JSON (Cui et al., 2025a; Li et al., 2025; Wei et al., 2025). This paradigm shift has broadened the scope of document parsing from recognizing isolated elements to jointly modeling content, structure, and reading order within a unified framework.

Despite the recent paradigm shift, most traditional and recent MLLM-based methods still follow an explicit or implicit "Crop–OCR–Merge" recipe (Cui et al., 2025a; Li et al., 2025; Niu et al., 2025). Even when document parsing is cast as a generative task, high-resolution pages are often handled by cropping the page with layout analysis, running OCR locally, and then merging the pieces into a final Markdown/JSON output. Such factorization is effective but prone to error propagation, where early mistakes are amplified during aggregation and hurt global consistency or fine-grained text fidelity (Kim et al., 2022; Wei et al., 2024). This motivates a natural question: *Is it possible to learn an end-to-end model that directly parses full-page inputs into a structured format, such as Markdown, without relying on split-and-merge pipelines, while still recovering accurate content while maintaining reading order?*

An end-to-end parser must overcome several obstacles. First, real-world documents vary widely in templates, domains, and visual styles (Wang et al., 2025a; Hui et al., 2024; Feng et al., 2025). Although existing OCR and parsing systems are strong, converting their outputs into high-quality, structure-preserving supervision is far from trivial: annotations may be incomplete, misordered, or internally inconsistent, and such noise is further amplified when training on long, structured sequences, particularly for tables and formulas. Second, a single page often interleaves dense text, complex tables, mathematical formulas, and recurring categories such as headers and footers (Pfitzmann et al., 2022; Blecher et al., 2023). Effective parsing therefore requires the model to master each constituent while still producing a coherent, reading-order-consistent serialization of the entire document, rather than improving one aspect at the expense of another.

To address these challenges, we propose OvisOCR, a strictly end-to-end document parser that aligns specialized perception with general reasoning in both data construction and post-training. Specifically, we alleviate the supervision bottleneck by casting label generation as a cross-model distillation process: a strong OCR engine supplies dense, fine-grained candidates, while a general-purpose model audits global consistency—e.g., reading-order coherence and format validity—and filters out annotations that are locally plausible yet globally inconsistent. On the optimization side, OvisOCR tackles the tension between token-level fidelity and global structural consistency via decomposed credit assignment. Building on an existing post-training recipe, we employ element-specific rewards for heterogeneous constituents (text/table/formula) and category-wise batch-normalized advantages, providing targeted learning signals for each part while still optimizing a single coherent, readable serialization. Experimental results on OmniDocBench v1.5 (Ouyang et al., 2025) show that OvisOCR achieves the best overall parsing performance and markedly improves in text fidelity and reading order. Our contributions are threefold:

- Strictly end-to-end parsing. We present OvisOCR, a compact (1B) strictly end-to-end document parser that maps full-page images directly to structured Markdown/JSON, eliminating Crop–OCR–Merge cascades.
- Principled training and optimization. We develop a training recipe that combines OCR-bootstrapped supervision with general-model consistency auditing and applies post-training with element-specific rewards and category-wise advantage normalization.
- Strong empirical performance. On OmniDocBench, OvisOCR achieves the best Overall score among compared methods and delivers notable gains in text fidelity, formula recognition, and reading order, validating compact end-to-end parsing at scale for document parsing.

# 2. Related Work

In this section, we review prior work along two lines: (i) traditional OCR systems, which are typically built as modular pipelines for text recognition and document understanding, and (ii) MLLMs for document parsing, which reframe parsing as a unified vision-to-structure generation problem.

## 2.1. Traditional OCR Systems

Traditional approaches to document parsing generally adopt a modular "divide-and-conquer" pipeline, decomposing the complex task into a series of specialized sub-problems: text detection, text recognition, layout analysis, and logical structure recovery (Wang et al., 2024; Li et al., 2025). Early systems relied on handcrafted features and heuristic rules, which often struggled with varying document templates. In the deep learning era, these components have evolved into robust neural networks. Text detection models like DB-Net (Liao et al., 2020) and recognition backbones such as CRNN (Shi et al., 2016) and SVTR (Du et al., 2022) have become standard components, offering high precision for localized text regions. To recover the document structure, these systems typically employ a separate layout analysis module to classify and localize regions before organizing them into a coherent sequence.

Several open-source frameworks have integrated these modules into powerful parsing tools. PaddleOCR (Cui et al., 2025a), for instance, constructs a comprehensive system by combining lightweight detection and recognition models with specialized modules for table structure recognition (TSR) and layout analysis. Similarly, MinerU (Wang et al., 2024; Niu et al., 2025) and Marker (Vik, 2025) employ high-precision layout detection to segment documents into blocks, which are then routed to dedicated recognizers for text, formulas, or tables. While these "Crop-OCR-Merge" pipelines achieve high accuracy on standard benchmarks, they face inherent limitations. The strict dependency between modules leads to error propagation, where a missed detection or incorrect layout classification inevitably corrupts the final output. Furthermore, the reassembly of recognized content often relies on spatial heuristics, which lack semantic awareness and frequently fail to recover the correct reading order in complex, multi-column layouts.

## 2.2. MLLMs for Document Parsing

General-purpose MLLMs (Team et al., 2023; Chen et al., 2024; OpenAI, 2024; Bai et al., 2025; Lu et al., 2025; Wang et al., 2025c; Jiulong et al., 2026; Kunming et al., 2026) exhibit impressive zero-shot and few-shot parsing capabilities due to their extensive pre-training. Concurrently, a new line of research focuses on developing models specifically tailored for the document domain (Wei et al., 2024; Agrawal et al., 2024; Poznanski et al., 2025; Feng et al., 2025; Nanonets, 2025; Jia et al., 2026; Hao et al., 2026; Yaoling et al., 2026). Early works such as Donut (Kim et al., 2022) and Nougat (Blecher et al., 2023) pioneered this direction by using encoder-decoder architectures to map images directly to sequences, though they were limited by low input resolution and smaller language backbones.

Recently, the field has witnessed a surge of specialized MLLMs that push the boundaries of resolution and accuracy. MonkeyOCR (Li et al., 2025) leverages a Structure-Recognition-Relation (SRR) triplet paradigm, which simplifies a complex multi-tool pipeline and avoids the inefficiencies of processing full pages with giant end-to-end models. The dots.ocr (RedNote, 2025) is good at multilingual document parsing and unifies layout detection and content recognition within a single vision-language model while maintaining reading order. PaddleOCR-VL (Cui et al., 2025a) introduces a compact architecture integrating a NaViT-style dynamic resolution encoder with the LLM, claiming SOTA performance by optimizing resource consumption. Although HunyuanOCR (Team et al., 2025) proposes a lightweight purely end-to-end model that eliminates layout dependencies, utilizing an MLP adapter to connect a native ViT with an LLM, the result is very difficult to reproduce.

Despite these advancements, challenges persist in two key areas. First, methodologically, many high-performance models still rely on implicit or explicit image slicing to handle detailed content, which can disrupt global layout coherence and mimic the "Crop-Merge" drawbacks of traditional pipelines (Cui et al., 2025a; ChatDOC, 2025; AI, 2025). Second, from an optimization perspective, training a single model to handle heterogeneous elements (text, tables, formulas) often leads to objective conflicts: improving high-level Markdown formatting can degrade fine-grained character recognition, and vice versa. Unlike prior works that rely primarily on architecture tweaks or data scaling, OvisOCR introduce a strictly slicing-free, holistic architecture to explicitly align the model's capabilities, ensuring that structural reasoning does not compromise perceptual fidelity.

## 3. OvisOCR

In this section, we present the OvisOCR framework, as shown in Figure 2, designed to achieve high-fidelity document parsing through a strictly end-to-end paradigm. Unlike traditional systems that physically isolate perception and structuring modules, OvisOCR internalizes these capabilities into a unified model, addressing the core challenge of aligning specialized perception (fine-grained text recognition) with general reasoning (structural and semantic coherence) through two synergistic pillars: synergized data construction and multi-grained alignment.

### 3.1. Synergized Data Construction

The scarcity of high-quality, diverse document-to-Markdown training data constitutes a primary bottleneck for end-to-end parsing. Current datasets typically rely on synthetic rendering, which inherently suffers from limited visual variability, or heuristic rule-based conversion, which frequently introduces severe formatting inconsistencies (Li

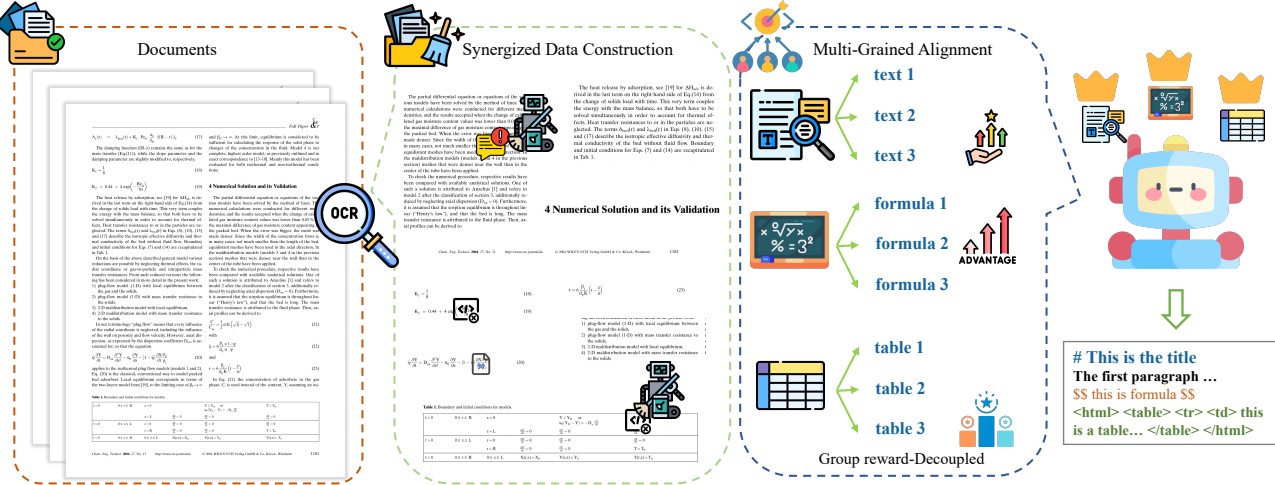

*Figure 2.* The overview of OvisOCR.

et al., 2025; RedNote, 2025). To address this limitation, we propose a cross-model distillation pipeline that generates high-fidelity supervision by synthesizing the complementary strengths of two distinct experts: a *Specialized Perceiver* and a *General Reasoner*.

**Local Evidence Extraction.** We first employ a mature, specialized OCR engine to extract fine-grained visual signals. For a given document image $\mathbf{I}$, this perception module outputs a set of unstructured candidates $\mathcal{C} = \{(b_i, c_i, t_i)\}_{i=1}^{N}$, where $b_i$ denotes the spatial bounding box, $c_i$ represents the element category, and $t_i$ contains the recognized content.

Crucially, the category $c_i$ encompasses a diverse range of semantic types, such as text, table, formula, header, and footer. The content $t_i$ adapts to the specific category: for table elements, $t_i$ preserves the internal structure via HTML sequences; whereas for image elements, we retain only the bounding box $b_i$ to indicate visual placement.

To construct a preliminary input sequence, following PaddleOCR-VL (Cui et al., 2025a), we serialize these candidates into a pseudo-Markdown format using a deterministic geometric sorting strategy, where bounding boxes $b_i$ are arranged primarily by vertical position and secondarily by horizontal alignment. While this spatial linearization effectively captures the raw textual content, the resulting reading order is frequently unreliable for complex layouts such as multi-column papers or sidebars due to the absence of semantic awareness. Nevertheless, this sequence provides high-recall local evidence, serving as a robust "rough draft" that effectively constrains the search space for the subsequent reasoning stage.

**Semantic Auditing and Filtering.** While the spatially serialized candidates provide a preliminary content baseline, the coordinate-based reading order is frequently unreliable, and the raw output often lacks semantic coherence. To elevate this noisy intermediate representation into high-quality training supervision, we introduce a Semantic Auditing stage.

We leverage a general MLLM to assess and verify the structural integrity of the parsed content. Crucially, we avoid simply prompting the MLLM to "fix" the text, as unrestricted generation often introduces severe issues, such as hallucinations or the omission of dense textual regions. Instead, we design a rigorous quality assurance protocol to strictly filter the dataset. As detailed in our prompt design (see Appendix A), we configure the MLLM to act as an "Expert OCR Quality Assurance Auditor." In this role, the model accepts the document image and the candidate Markdown sequence as paired inputs and executes a multi-dimensional verification based on the following criteria:

1. **Hallucination Detection:** To mitigate the issue of autoregressive generation loops common in general MLLMs (Jafari & Anbarjafari, 2025; Dong et al., 2025), the auditor identifies repetitive text patterns. It strictly differentiates between model-induced hallucinations and legitimate visual repetitions, such as empty tables or repeated long numbers.

2. **Content Completeness:** The protocol evaluates the overall semantic quality to filter out degenerate outputs. This involves flagging instances that are either semantically vacuous, containing meaningless characters or text in images, or those that suffer from significant content truncation compared to the visual input.

3. **Syntactic Validity:** For documents containing heterogeneous elements, simple text matching is insufficient. The auditor enforces strict formatting constraints: it verifies that the topological structure of HTML tables aligns visually with the image layout and ensures that mathematical formulas adhere to valid syntax.

4. **Logical Reading Order:** Crucially, the protocol validates the inter-category sequencing. It checks whether the serialization of diverse constituents follows the visual logical flow, identifying errors where the spatial heuristic might have misordered multi-column content or sidebars.

The auditor outputs a structured JSON object containing boolean flags for each error type. We strictly filter out any samples flagged with errors, ensuring that the final dataset, which is constructed via this cross-model distillation, possesses both the fine-grained character accuracy of the specialized OCR engine and the structural correctness and reading order verified by the general reasoning model.

### 3.2. Multi-Grained Alignment

End-to-end document parsing requires optimizing heterogeneous content types—specifically text, tables, and formulas—which often exhibit conflicting learning dynamics. Standard reinforcement learning approaches frequently suffer from optimization imbalances, where the model prioritizes easier modalities at the expense of others. To resolve this, we implement a decomposed credit assignment strategy. Drawing inspiration from Group reward-Decoupled Normalization Policy Optimization (GDPO) (Liu et al., 2026), we extend its reward-decoupling philosophy to the document parsing domain.

**Revisiting GRPO and Reward Collapse.** Group Relative Policy Optimization (GRPO) (Shao et al., 2024) has emerged as a resource-efficient alternative to PPO, eliminating the need for a value function by estimating advantages via group-wise normalization. Formally, for a prompt $q$ and a group of outputs $\{o_i\}_{i=1}^{G}$ sampled from the old policy $\pi_{\theta_{old}}$, standard GRPO computes the advantage $A_i$ by normalizing the *aggregated* reward $r_{\text{total}} = \sum_k r_k$:

$$A_i = \frac{r_{\text{total}}(o_i) - \text{mean}(\{r_{\text{total}}(o_j)\})}{\text{std}(\{r_{\text{total}}(o_j)\}) + \epsilon} \qquad (1)$$

However, recent studies (Liu et al., 2026) identify a critical limitation termed Reward Collapse. When summing heterogeneous rewards in GRPO (e.g., text accuracy vs. format validity) before normalization, distinct combinations of successes and failures can map to identical advantage values. This loss of resolution prevents the model from distinguishing between "correct content but wrong format" and "correct format but wrong content," leading to suboptimal convergence where the model often sacrifices perceptual detail for structural formatting. Moreover, the document parsing task entails faithful reproduction rather than complex reasoning; the primary conflict arises not from thinking format but from the heterogeneity of content categories.

**GDPO in Document Parsing.** To mitigate the interference between these heterogeneous elements, we extend the GDPO framework. Unlike standard GRPO that normalizes the aggregated sum, GDPO normalizes the reward component $r_k$ for each component independently within the group to obtain component-wise advantages $A_i^{(k)}$, and then aggregates them:

$$A_i^{\text{GDPO}} = \sum_k \frac{r_k(o_i) - \text{mean}(\{r_k(o_j)\})}{\text{std}(\{r_k(o_j)\}) + \epsilon} \qquad (2)$$

This decomposed advantage assignment ensures that the optimization landscape for each content category remains independent. It effectively prevents the model from neglecting difficult elements (e.g., formulas and tables) simply because other elements (e.g., plain text) yield higher raw scores, thereby forcing concurrent improvement across all distinct document constituents.

**Category-Specific Reward Design.** Under this decoupled framework, we are able to design a set of category-specific reward functions tailored to the unique structural properties of document parsing. Rather than optimizing a single coarse page-level metric, we parse the generated Markdown sequences and ground truth into constituent blocks to compute targeted fidelity scores:

- **Text Fidelity Reward ($\mathcal{R}_{\text{text}}$):** To enforce strict character-level accuracy, we define the reward based on the Normalized Edit Distance (NED). Specifically, we calculate $\mathcal{R}_{\text{text}} = 1 - \text{NED}(\hat{y}, y)$, where $\hat{y}$ and $y$ denote the predicted and ground-truth text segments, respectively. This formulation explicitly penalizes hallucinations and omissions, serving as the primary anchor for perceptual precision.
- **Table Structure Reward ($\mathcal{R}_{\text{table}}$):** For tables, we adopt the Tree-Edit-Distance-based Similarity (TEDS) (Zhong et al., 2020) score. TEDS holistically evaluates both the structural topology within `<tr>`, `<td>`, and the corresponding cell content, encouraging the model to generate valid and aligned table representations.
- **Formula Visual Reward ($\mathcal{R}_{\text{formula}}$):** For mathematical expressions, we employ the Character Detection Matching (CDM) (Wang et al., 2025b). Unlike text-based metrics, CDM renders latex code into images for visual matching. This mitigates ambiguity from diverse notations, rewarding visual correctness rather than specific syntax.

By integrating these signals for advantages, OvisOCR effectively harmonizes the optimization of fine-grained character recognition with complex structural reconstruction. This extension ensures concurrent improvements across complex documents, yielding a robust end-to-end parser that maintains both perceptual precision and structural integrity.

### 3.3. Training Recipe

To effectively internalize the diverse capabilities required for document parsing—ranging from fine-grained character recognition to complex layout reconstruction—we adopt a progressive two-stage training curriculum. This strategy transitions from supervised knowledge injection to preference-based structural alignment.

**Model Architecture**. OvisOCR is built upon the Ovis architecture (Lu et al., 2024; 2025). It utilizes a high-resolution visual encoder SigLIP2-400m to compress document images into visual tokens, which are then projected into the embedding space of a compact LLM Qwen3-0.8B via a textual embedding table. This lightweight choice ensures efficient inference without compromising learing capabilities. The specialized perceiver we used in 3.1 is PaddleOCR-VL, and the general reasoner is Gemini2.5-pro (Team et al., 2023).

**Stage 1: Foundational Supervised Fine-Tuning (SFT).** The primary goal of this stage is to establish the model's fundamental capability to map visual features to valid Markdown syntax and follow the logical reading order. We perform full-parameter fine-tuning using the high-quality dataset as constructed in 3.1.

To ensure the model excel in document parsing, we construct various the document-parsing data without mix general data to construct a specialized document parsing model. We train the model for only 1 epoch with a global batch size of 512. We employ a standard auto-regressive cross-entropy loss. The learning rate is warmed up to $5e-6$ and then keep constant. To handle high-resolution inputs typical in documents, we set the maximum context length to 16384 tokens, utilizing dynamic resolution techniques to preserve visual details.

**Stage 2: Structure-Aware Preference Alignment.** While SFT establishes basic competence, models often struggle with structural details, such as aligning complex table rows or distinguishing similar mathematical symbols, due to the lack of explicit negative feedback in standard likelihood maximization. In this stage, we extend the GDPO algorithm to learn from various contents, aligning the model with the multi-grained rewards defined in 3.2.

For each input image, we sample 8 different rollouts to cover a diverse space of potential parsing results. This allows the model to explore different formatting strategies, such as various LaTeX representations for the same formula, and learn which best satisfies the structural constraints. We train the model for 2 epoch with a global batch size of 256. The learning rate is warmed up to $5e-7$ and then follows a cosine decay schedule. This ensures sustained exploration and stable gradient updates. We set the KL-divergence coefficient $\beta$ to $0.2$ to prevent the policy from deviating too far from the SFT initialization, ensuring that the parsing fluency gained in Stage 1 is preserved while structural precision is refined.

## 4. Experiments

In this section, we conduct a comprehensive empirical evaluation to validate the effectiveness and efficiency of OvisOCR. We begin by detailing the experimental setup, a holistic comparison against a wide range of baselines, and finally, quantitative analysis and qualitative case studies.

### 4.1. Experimental Setup

**Datasets.** OvisOCR is a specialized document parsing model whose goal is to directly transform a full-page document image into a structured Markdown representation. It is not intended to serve as a general-purpose instruction-following MLLM. Instead, this work focuses on high-fidelity end-to-end parsing, including text transcription, formula recognition, table reconstruction, and reading-order preservation. Although document parsing is only one component of document intelligence, it provides the essential perceptual foundation for downstream understanding and reasoning tasks. Therefore, our objective is to demonstrate that a compact end-to-end parser can outperform complex Crop-OCR-Merge pipelines, thereby simplifying the path toward future document intelligence systems.

Therefore, we conduct our primary evaluations on OmniDocBench v1.5 (Ouyang et al., 2025), a comprehensive benchmark consisting of 1,355 diverse document pages. This dataset is selected for its high linguistic diversity (balanced Chinese/English) and structural complexity, featuring dense mathematical formulas, intricate tables, and varying layouts. These characteristics make it a rigorous testbed for document parsers in complex real-world scenarios.

**Evaluation Metrics.** Following standard protocols, we assess performance using a multi-grained approach covering distinct element types: (1)Text: Evaluated using standard Edit Distance to measure character-level fidelity. (2)Formulas: Assessed via the Character Detection Matching (CDM) (Wang et al., 2025b) metric to ensure LaTeX syntactic correctness. (3)Tables: Evaluated using both Tree-Edit-Distance-based Similarity (TEDS) (Zhong et al., 2020) and its structural variant TEDS-Structure to evaluate the stricture alignment.

To provide a holistic view, we report an Overall score, calculated as the average of the Text, Formula (CDM), and Table (TEDS) scores. Additionally, Reading Order is also included to evaluate serialization logic, though this metric is not included in the Overall calculation. Beyond aggregate scores, we also provide fine-grained performance breakdowns across different dimensions, such as specific

document types, to offer a comprehensive assessment.

## 4.2. Main Results and Analysis

**Performance Analysis.** As presented in Table 1, OvisOCR achieves an Overall score of 93.19 on OmniDocBench v1.5, ranking first among all compared methods. This result demonstrates the effectiveness of our end-to-end method across diverse document parsing paradigms, including specialized MLLMs, general MLLMs, and pipeline tools.

*OvisOCR surpasses Specialized MLLMs.* OvisOCR achieves the best Overall score among specialized MLLMs, outperforming the highly optimized PaddleOCR-VL as well as larger specialized models. Notably, OvisOCR not only surpasses DeepSeek-OCR-2 with fewer parameters, but also improves over PaddleOCR-VL in Overall score, text edit distance, formula recognition, and reading order. Despite its compact scale, OvisOCR achieves higher accuracy in both text edit distance and formula recognition, indicating that our proposed training recipe, which combines cross-model distillation with GDPO preference alignment, effectively activates the model's potential in fine-grained perception and structured document parsing.

*Better Efficiency against General MLLMs.* A striking observation is OvisOCR's efficiency compared to massive general-purpose models. Despite utilizing only 1B parameters, OvisOCR outperforms giants like Qwen3-VL-235B and Gemini-2.5 Pro by substantial margins around 3 points. These general models, while powerful in reasoning, often struggle with the pixel-level fidelity required for document parsing, exhibiting significantly higher text error rates. This validates our hypothesis that specialized alignment is crucial for document parsing and that simple scaling laws are insufficient to solve the nuances of fine-grained OCR.

*Dominance over Pipeline Tools.* Traditional "Crop-OCR-Merge" pipelines lag significantly behind end-to-end approaches. Even the robust PP-StructureV3 underperforms OvisOCR by nearly 5 points, while open-source tools like Marker score only 71.30. The advantage of OvisOCR's end-to-end paradigm is most evident in the Reading Order metric. Pipeline methods often fail to serialize complex layouts correctly, e.g., Marker's Reading Order edit distance is 0.250, nearly 6 times worse than OvisOCR's 0.044. In contrast, OvisOCR maintains global coherence comparable to the best-performing models, proving that holistic modeling eliminates the error propagation during split-and-merge.

## 5. Analysis

In this section, we first perform ablation studies to isolate the individual contributions of our components. We then evaluated the computational efficiency of our end-to-end OvisOCR compared to baselines. We also provide a fine-grained analysis of the model's performance and qualitative visualizations to demonstrate OvisOCR's capabilities in handling complex real-world documents in Appendix B.

## 5.1. Ablation Study

We conduct a comprehensive ablation study to isolate the effects of the reward mechanism, data quality, and normalization strategy. As shown in Figure 3, the SFT baseline achieves an Overall score of 92.30. Applying GRPO with a single global Markdown-level edit-distance reward (Ovis-GRPO-MD) improves the score to 92.35, but the gain is relatively limited. This indicates that a coarse page-level reward cannot sufficiently capture the heterogeneous error patterns in document parsing. Since text, formulas, and tables differ substantially in both representation and evaluation criteria, aggregating them into a single reward may cause reward collapse, where the model mainly optimizes easier components while under-optimizing complex ones.

To address this issue, Ovis-GRPO-DR introduces decoupled rewards for different document constituents. This variant improves the Overall score to 92.84 and brings clear gains in Formula CDM and Table TEDS, confirming that category-specific rewards provide more precise credit assignment for fine-grained perception and structural reconstruction. We then evaluate the effect of data quality by applying the Semantic Auditor to filter noisy or globally inconsistent samples. Ovis-GRPO-SA further improves the Overall score to 93.07, demonstrating that high-quality audited data provides a more reliable optimization space for reinforcement learning. Finally, the full Ovis-GDPO model combines audited data with category-wise advantage normalization and achieves the best Overall score of 93.19. The consistent improvements across all major metrics validate the synergy between the data construction pillar and the multi-grained alignment pillar. Notably, Ovis-GDPO also surpasses PaddleOCR-VL in Overall score, suggesting that a compact end-to-end parser can outperform strong Crop-OCR-Merge systems when specialized perception and structural reasoning are properly aligned.

## 5.2. Inference Efficiency and Resource Costs

Table 2 shows that our end-to-end OvisOCR achieves the best overall inference efficiency among all compared methods under a unified vLLM backend on OmniDocBench v1.5. Specifically, OvisOCR-1B attains the lowest total runtime and the highest page throughput (pages/s), demonstrating the advantage of an end-to-end architecture for document parsing. This efficiency gain is largely attributed to the removal of repeated serialization, intermediate representations, and stage-wise coordination overhead that are commonly introduced by pipeline-based systems. While PaddleOCR-VL achieves the highest token throughput, Ovi-

*Table 1.* Comprehensive evaluation of document parsing on OmniDocBench v1.5 (Ouyang et al., 2025). Text$^E$ represents Text$^{Edit}$, Formula$^C$ represents Formula$^{CDM}$, Table$^T$ represents Table$^{TEDS}$, Table$^S$ represents Table$^{TEDS-S}$, RO$^E$ represents Reading Order$^{Edit}$. OvisOCR demonstrates the best Overall performance among all compared methods, surpassing specialized MLLMs, massive general MLLMs, and traditional pipeline tools, validating the effectiveness of our end-to-end alignment strategy.

| Model Type | Methods | Size | Overall↑ | Text$^E$↓ | Formula$^C$↑ | Table$^T$↑ | Table$^S$↑ | RO$^E$↓ |
|---|---|---|---|---|---|---|---|---|
| Specialized MLLMs | OvisOCR | 1B | **93.19** | **0.032** | **91.99** | 90.79 | 93.75 | **0.033** |
| | PaddleOCR-VL | 0.9B | 92.86 | 0.035 | 91.22 | **90.89** | **94.76** | 0.043 |
| | DeepSeek-OCR-2 | 3B | 91.09 | 0.048 | 90.31 | 87.75 | 92.06 | 0.057 |
| | MinerU2.5 | 1.2B | 90.67 | 0.047 | 88.46 | 88.22 | 92.38 | 0.044 |
| | MonkeyOCR-pro-3B | 3B | 88.85 | 0.075 | 87.25 | 86.78 | 90.63 | 0.128 |
| | OCRVerse | 4B | 88.56 | 0.058 | 86.91 | 84.55 | 88.45 | 0.071 |
| | dots.ocr | 3B | 88.41 | 0.048 | 83.22 | 86.78 | 90.62 | 0.053 |
| | MonkeyOCR-3B | 3B | 87.13 | 0.075 | 87.45 | 81.39 | 85.92 | 0.129 |
| | Deepseek-OCR | 3B | 87.01 | 0.073 | 83.37 | 84.97 | 88.80 | 0.086 |
| | MonkeyOCR-pro-1.2B | 1.2B | 86.96 | 0.084 | 85.02 | 84.24 | 89.02 | 0.130 |
| | Nanonets-OCR-s | 3B | 85.59 | 0.093 | 85.90 | 80.14 | 85.57 | 0.108 |
| | MinerU2-VLM | 0.9B | 85.56 | 0.078 | 80.95 | 83.54 | 87.66 | 0.086 |
| | olmOCR | 7B | 81.79 | 0.096 | 86.04 | 68.92 | 74.77 | 0.121 |
| | Dolphin-1.5 | 0.3B | 83.21 | 0.092 | 80.78 | 78.06 | 84.10 | 0.080 |
| | POINTS-Reader | 3B | 80.98 | 0.134 | 79.20 | 77.13 | 81.66 | 0.145 |
| | Mistral OCR | - | 78.83 | 0.164 | 82.84 | 70.03 | 78.04 | 0.144 |
| | OCRFlux | 3B | 74.82 | 0.193 | 68.03 | 75.75 | 80.23 | 0.202 |
| | Dolphin | 0.3B | 74.67 | 0.125 | 67.85 | 68.70 | 77.77 | 0.124 |
| General MLLMs | Qwen3-VL-235B-A22B | 235B | 89.15 | 0.069 | 88.14 | 86.21 | 90.55 | 0.068 |
| | Qwen3-VL | 32B | 88.45 | 0.057 | 83.71 | 87.34 | 91.03 | 0.069 |
| | Gemini-2.5 Pro | - | 88.03 | 0.075 | 85.82 | 85.71 | 90.29 | 0.097 |
| | Qwen2.5-VL | 72B | 87.02 | 0.094 | 88.27 | 82.15 | 86.22 | 0.102 |
| | InternVL3.5 | 241B | 82.67 | 0.142 | 87.23 | 75.00 | 81.28 | 0.125 |
| | GLM4.6V-flash | 9B | 82.25 | 0.233 | 87.80 | 82.25 | 86.80 | 0.349 |
| | InternVL3 | 78B | 80.33 | 0.131 | 83.42 | 70.64 | 77.74 | 0.113 |
| | GPT-4o | - | 75.02 | 0.217 | 79.70 | 67.07 | 76.09 | 0.148 |
| | Qwen3-VL-30B-A3B | 30B | 74.52 | 0.060 | 83.42 | 46.14 | 49.72 | 0.078 |
| Pipeline Tools | PP-StructureV3 | - | 86.73 | 0.073 | 85.79 | 81.68 | 89.48 | 0.073 |
| | Mineru2-pipeline | - | 75.51 | 0.209 | 76.55 | 70.90 | 79.11 | 0.225 |
| | Marker-1.8.2 | - | 71.30 | 0.206 | 76.66 | 57.88 | 71.17 | 0.250 |

sOCR offers the most favorable trade-off in practical end-to-end deployment due to its shorter wall-clock time and stronger page-level efficiency.

### 5.3. Generalization to More Benchmarks

We use OmniDocBench v1.5 as our primary evaluation benchmark because it provides a rigorous and comprehensive testbed for structured document parsing. Unlike clean OCR datasets, OmniDocBench covers diverse and challenging document scenarios, including dense academic pages, complex tables, mathematical formulas, highly stylized layouts, and newspaper-style documents. Therefore, the strong performance of OvisOCR on OmniDocBench already reflects its ability to handle complex and non-standard visual structures.

To further evaluate the generalizability and robustness of OvisOCR beyond OmniDocBench, we conduct additional experiments in Table 3. OvisOCR consistently outperforms existing specialized document parsing models across all three benchmarks.

These results demonstrate that the advantage of OvisOCR is not limited to OmniDocBench. Instead, the proposed end-to-end alignment strategy also generalizes well to broader OCR-oriented benchmarks. The consistent improvements indicate that OvisOCR possesses stronger text recognition and spatial reasoning capabilities, further validating the effectiveness of aligning specialized perception with general reasoning in a unified end-to-end parser.

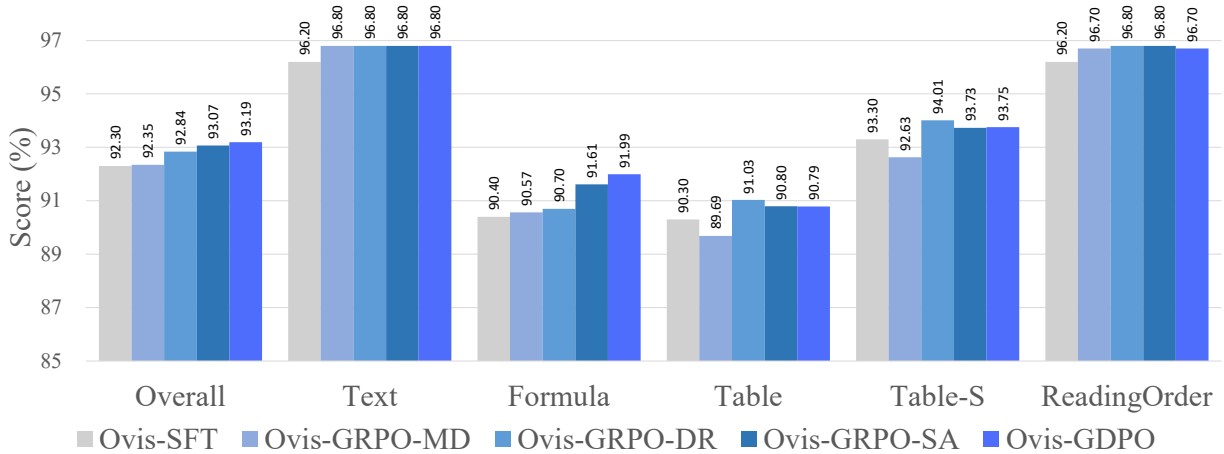

*Figure 3.* Ablation Study of OvisOCR. MD uses a single global markdown-level edit distance as the reward; DR means Decoupled Reward to specific categories (text, formula, and table); SA means Semantic Auditor with filtered high-quality data. By isolating the reward mechanism, the data quality, and the normalization strategy, we demonstrate that our gains are driven by a synergistic combination of both pillars: synergized data construction and multi-grained alignment.

*Table 2.* Inference performance comparison. All methods are tested under the vLLM backend on OmniDocBench v1.5. Our end-to-end OvisOCR model with 1B parameters achieves the best latency and page-level throughput.

| Methods | Total Time (s)↓ | Pages/s↑ | Tokens/s↑ |
|---|---|---|---|
| PaddleOCR-VL-0.9B (Cui et al., 2025a) | 781.29 | 1.734 | **2882.11** |
| MinerU2.5-1.2B (Niu et al., 2025) | 796.53 | 1.701 | 2425.69 |
| MonkeyOCR-pro-1.2B (Li et al., 2025) | 2273.30 | 0.596 | 927.34 |
| dots.ocr (RedNote, 2025) | 1169.94 | 1.158 | 1806.70 |
| OvisOCR-1B | **744.57** | **1.820** | 2639.34 |

*Table 3.* Generalization results on more OCR benchmarks. OvisOCR consistently outperforms existing specialized models across OCRBench, OCRBench v2-EN (v2E), and OCRBench v2-CN (v2C).

| Method | OCRBench | v2E | v2C |
|---|---|---|---|
| PaddleOCR-VL | 549 | 18.1 | 40.9 |
| MinerU2.5 | 331 | 17.1 | 40.0 |
| MonkeyOCR | 655 | 21.8 | 38.9 |
| dots.ocr | 625 | 19.6 | 35.9 |
| OvisOCR | **701** | **23.7** | **43.2** |

v1.5 demonstrate that our model achieves the best Overall performance among compared methods, outperforming strong specialized parsers, much larger general MLLMs, and traditional pipeline tools. Other results highlight that OvisOCR offers a favorable balance between parsing quality and inference efficiency. By removing cascaded layout detection, cropping, recognition, and merging stages, it reduces both engineering complexity and potential error propagation while maintaining strong performance. This streamlined end-to-end design turns OCR outputs into structured, downstream-ready representations, providing a solid interface for future document analysis systems.

## 6. Conclusion

In this paper, we presented OvisOCR, a lightweight and strictly end-to-end MLLM which directly maps full-page document images to structured Markdown. OvisOCR aligns specialized perception with general reasoning through (i) cross-model distillation for data construction, which bootstraps dense candidates from a strong OCR engine and filters them with a consistency auditor, and (ii) multi-grained optimization with element-aware credit assignment for text, tables, and formulas. Experimental results on OmniDocBench

## Acknowledgements

This work is partially supported by NSFC (62522605), Basic Research Program of Jiangsu under Grants (BK20253021), Natural Science Foundation of Jiangsu Province of China under Grant (BK20250062), the Fundamental and Interdisciplinary Disciplines Breakthrough Plan of the Ministry of Education of China (No. JYB2025XDXM118), the "111 Center" (No. B26023), the Collaborative Innovation Center of Novel Software Technology and Industrialization.

## Impact Statement

This paper presents work whose goal is to advance the field of Machine Learning, especially in Document Parsing. There are many potential societal consequences in our work related to our model and benchmark, none of which we feel must be specifically highlighted here.

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

# A. Details for Synergized Data Construction

We use the following prompt to leverage a general MLLM to assess and verify the structural integrity of the parsed content.

---

You are an expert OCR Quality Assurance Auditor.
Your task is to verify the quality of the OCR Markdown Output against the provided Image.
**Input Markdown to Verify:**
markdown text
**Evaluation Criteria:**
1. **Empty/Garbage**: Is the content meaningful?
2. **Repetition (Hallucination)**: Does the model generate repetitive loops (e.g., "the the the", repeating the same sentence 3+ times) that DO NOT exist in the image?
*Note: If the image actually has repeated text (like a decorative border), it is NOT an error.*
3. **Table Structure**: If there are tables, is the HTML structure ('<table>') valid and does it visually align with the image content?
4. **Formulas**: If there are math formulas, are they correctly formatted in LaTeX and syntactically valid?
5. **Overall Text**: Are there significant missing chunks or completely wrong text?
**Output Format:**
Return a strictly valid JSON object with boolean flags:
{
"has_content": boolean, // True if text was extracted, False if empty/null
"is_repetitive_hallucination": boolean, // True if model is looping/repeating text NOT in image
"is_table_error": boolean, // True if table HTML is broken or content is severely misaligned
"is_formula_error": boolean, // True if LaTeX is broken
"is_text_error": boolean, // True for garbage output or major missing sections
"reasoning": "Short explanation of the flags set to true."
}

---

The auditor outputs a structured JSON object containing boolean flags for each error type. We strictly filter out any samples flagged with errors, ensuring that the final dataset, which is constructed via this cross-model distillation, possesses both the fine-grained character accuracy of the specialized OCR engine and the structural correctness and reading order verified by the general reasoning model.

After filtering, we use the following prompt for OvisOCR to parse the document into the Markdown format.

---

Extract all readable content from the image in natural human reading order and output the result as a single Markdown document. For charts or images, represent them using an HTML image tag: ¡img src="images/bbox_left_top_right_bottom.jpg" />, where left, top, right, bottom are bounding box coordinates scaled to [0, 1000). Format formulas as LaTeX. Format tables as HTML: <table>...</table>. Transcribe all other text as standard Markdown. Preserve the original text without translation or paraphrasing.

---

# B. More Experiments and Analysis

### B.1. Fine-Grained Capability Assessment

In the main paper in Table 1, we report results from the official OmniDocBench evaluation. Here, we further analyze performance by data source; however, since the official source-level breakdown is not publicly available, we report results from our local evaluation instead.

As shown in Table 4, OvisOCR achieves top-tier performance comparable to PaddleOCR-VL across all three metrics: it is slightly better on Text Edit Distance and Formula CDM, while it still lags behind on Table TEDS. Notably, for text recognition, OvisOCR performs particularly well on research reports, magazines, and academic literature, indicating stronger robustness to long-form and academic-style layouts. For formula recognition, OvisOCR consistently outperforms PaddleOCR-VL on every source category where formula annotations are available, suggesting more reliable recovery of fine-grained mathematical symbols and formatting. Meanwhile, table reconstruction remains the primary gap, motivating future improvements for complex table structures and alignment accuracy.

*Table 4.* Detailed analysis by different data sources.

| | book | PPT2PDF | research_report | colorful_textbook | exam_paper | magazine | academic_literature | note | newspaper | mean |
|---|---|---|---|---|---|---|---|---|---|---|
| *Text Edit Distance* | | | | | | | | | | |
| PaddleOCR-VL | 0.031 | 0.034 | 0.031 | 0.088 | 0.048 | 0.037 | 0.026 | 0.060 | 0.037 | 0.043 |
| OvisOCR | 0.038 | 0.028 | 0.006 | 0.060 | 0.046 | 0.015 | 0.011 | 0.055 | 0.064 | 0.038 |
| *Table TEDS* | | | | | | | | | | |
| PaddleOCR-VL | 94.30 | 89.80 | 95.60 | 95.50 | 93.00 | 80.70 | 85.00 | 79.00 | 78.70 | 90.50 |
| OvisOCR | 90.60 | 82.30 | 93.80 | 94.30 | 93.80 | 68.50 | 83.40 | 77.60 | 64.50 | 87.90 |
| *Formula CDM* | | | | | | | | | | |
| PaddleOCR-VL | 89.20 | 82.10 | - | 92.10 | 89.60 | - | 94.30 | - | - | 88.50 |
| OvisOCR | 90.90 | 87.20 | - | 98.00 | 90.90 | - | 97.10 | - | - | 91.00 |

## B.2. Case Studies

Figure 4 presents representative cases where OvisOCR improves over pipeline-style parser PaddleOCR-VL on fine-grained details. In code snippets, small token-level deviations, such as confusing getContext with context, change the semantics of the program and cannot be reliably repaired after merging. Similarly, in mathematical expressions, subtle notation differences, e.g., $N$ vs. $\mathbb{N}$, carry precise meaning and are easy to miss when regions are processed independently. Across these cases, OvisOCR's strictly end-to-end generation from full-page inputs better preserves character- and symbol-level fidelity, producing more accurate Markdown serialization for both detailed text and formulas. This qualitative evidence complements our quantitative gains on OmniDocBench, and underscores the practical advantage of end-to-end document parsing when correctness hinges on small yet consequential details.

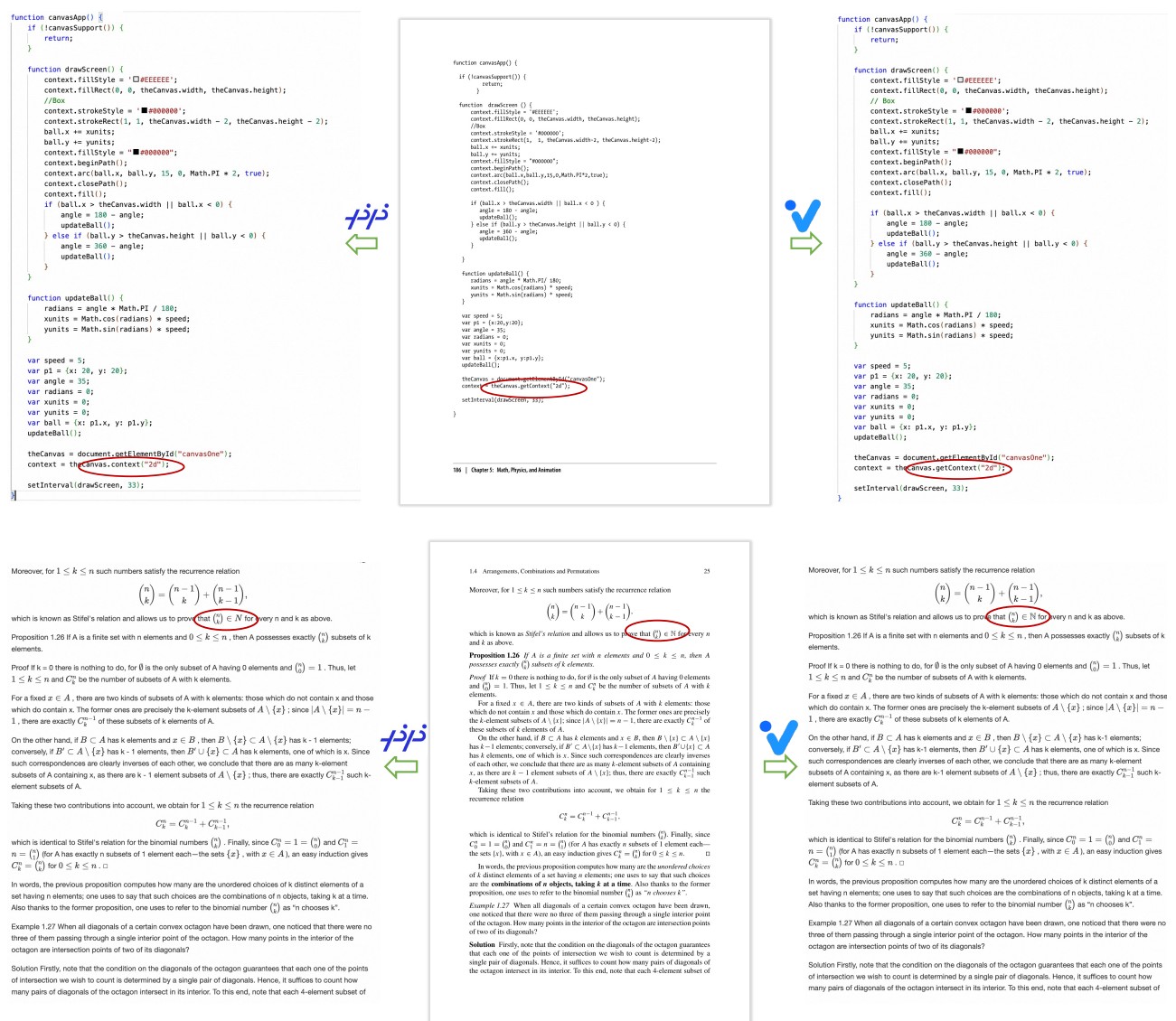

*Figure 4.* Case study highlighting OvisOCR's fine-grained accuracy of end-to-end parsing. (Top) In code regions, PaddleOCR-VL mis-recognizes precise API tokens theCanvas.getContext("2d") as theCanvas.context("2d"), while OvisOCR preserves the correct identifier in the final Markdown. (Bottom) In mathematical text, pipeline outputs confuse subtle but critical symbols, e.g., $N$ vs. $\mathbb{N}$, whereas OvisOCR correctly recovers the intended notation and yields a reading-consistent serialization.

