# OpenReview forum: "OvisOCR: End-to-End Document Parsing via Aligning Specialized Perception with General Reasoning"
_ICML.cc/2026/Conference — ICML 2026 regular_

### Official Review · Reviewer_ovQC · 2026-03-11

**Soundness:** 3
**Presentation:** 3
**Significance:** 2
**Originality:** 2
**Overall Recommendation:** 4
**Confidence:** 5

**Summary:**

This paper presents OvisOCR, a lightweight end-to-end Multimodal Large Language Model (MLLM) tailored for document parsing. To address the error accumulation and structural inconsistency inherent in traditional "Crop-OCR-Merge" pipelines, the authors propose a framework that aligns specialized perception (fine-grained text recognition) with general reasoning (structural and semantic coherence). The approach features a synergized data construction pipeline where a general MLLM serves as a "Semantic Auditor" to refine and validate the outputs of OCR engines, ensuring high-fidelity training data. Furthermore, the study introduces Category-Specific Rewards and a Group-based Decomposed Policy Optimization (GDPO) strategy to mitigate optimization imbalances across diverse document elements like text, tables, and formulas. Experimental results on the OmniDocBench benchmark demonstrate that OvisOCR, with only 2B parameters, achieves top-tier performance and high efficiency.

**Compliance With Llm Reviewing Policy:**

Affirmed.

**Final Justification:**

I appreciate the authors for their detailed rebuttal, which effectively clarified my concerns regarding the conceptual framework and experimental details. However, after carefully reconsidering the overall soundness and noverlty of the paper , I maintain my original score.

**Key Questions For Authors:**

1. Could the authors provide more specific details regarding the composition of the training dataset? Specifically, what is the total scale of the data (e.g., number of pages), and where do these documents come from?
2. The manuscript suggests that the Semantic Auditor primarily functions as a high-quality data refiner during the SFT (Supervised Fine-Tuning) stage. However, the current ablation studies seem to focus predominantly on its role during the RL (Reinforcement Learning) phase. Could the authors provide a quantitative analysis of the SA’s contribution specifically within the SFT stage?
3. While the paper introduces GDPO to mitigate optimization imbalances, it lacks a direct comparison with other RL optimization strategies. Could the authors provide experimental results comparing GDPO against other reinforcement learning methods, such as GRPO？

**Limitations:**

yes

**Strengths And Weaknesses:**

Strengths

Soundness:
The paper presents a rigorous and comprehensive empirical evaluation on the OmniDocBench benchmark. OvisOCR achieves top-tier performance not only in overall metrics but also across diverse and challenging document elements, including complex tables and mathematical formulas. Furthermore, the experimental results highlight that OvisOCR attains superior inference efficiency compared to existing baselines, underscoring its practical advantages for latency-sensitive and real-world deployment scenarios.

Originality:
The introduction of Group-based Decomposed Policy Optimization (GDPO) and Category-Specific Rewards within the reinforcement learning stage for document parsing is a commendable contribution. By effectively addressing optimization conflicts between heterogeneous document elements, this approach offers a fresh perspective that could inspire future research.

Presentation:
The manuscript is well-structured and maintains a clear, logical narrative. The technical components, from data construction to the optimization framework, are presented in a manner that is easy to follow and facilitates a solid understanding of the methodology.


Weaknesses

Soundness:
Several critical details are missing. Specifically, the paper lacks explicit information concerning the scale and source of the training datasets. Additionally, the ablation studies are incomplete: there is no quantitative analysis of the Semantic Auditor’s specific contribution during the SFT stage. Most importantly, the authors fail to provide a direct comparison between GDPO and standard RL algorithms (e.g., GRPO), making it difficult to evaluate the value of the optimization strategy.

Significance:
While the end-to-end MLLM approach for document parsing is effective, it is not a fundamentally new concept. Existing works such as OCRFlux and olmOCR have already explored similar architectures. Given this prior literature, the conceptual leap provided by this work appears incremental, which somewhat limits its overall impact on the field.

Originality:
The technical novelty is somewhat constrained. The model architecture follows standard MLLM designs without significant structural innovation. Furthermore, the GDPO is an off-the-shelf RL technique, and the Category-Specific Rewards used for different document element types, such as TEDS for tables and CDM for formulas, have been defined in previous works. Similarly, the idea of leveraging an LLM or MLLM for data quality filtering is a well-established practice in the industry (e.g., as seen in MinerU2.5), making the proposed framework less distinctive.

---

> ### Author Rebuttal · Authors · 2026-03-31
>
> We appreciate your recognition of our comprehensive evaluation, method, and writing. We will answer the questions below, and we hope this clears up your concerns.
>
> **Q1:** Could the authors provide more specific details regarding the composition of the training dataset?
>
> **A1:** We thank the reviewer for the request for more specific details regarding our data pipeline.
>
> For the SFT stage, OvisOCR was trained on a large-scale corpus of approximately 4 million high-quality samples (pages). This scale ensures that the model internalizes the foundational mapping between complex visual features and structured Markdown syntax across a wide variety of domains.
>
> To ensure robust generalization, we curated the dataset from a blend of high-traffic open-source repositories, academic archives, and specialized proprietary sources. The composition includes: arXiv papers, technical reports, and academic journals. We also sampled a diverse range of Chinese and English newspapers, magazines, and corporate financial reports, which feature varied layouts, sidebars, and diverse fonts. We also collect educational and instructional pdfs like textbooks, lecture PPTs, and educational materials. To handle "in-the-wild" document challenges, we supplemented the corpus with purchased datasets, including handwritten notes, digitized exam papers, and archived historical documents.
>
> For the RL stage, we selected a high-fidelity subset of verified 45k samples to focus the model on resolving the most challenging structural and perceptual conflicts with high quality.
>
> **Q2:**  Question about Semantic Auditor and Comparison with GRPO
>
> **A2:** We thank the reviewer for the opportunity to clarify the strategic role of the Semantic Auditor within our training pipeline.
>
> We would like to clarify that the **Semantic Auditor (SA) is primarily utilized during the RL stage** rather than the SFT stage. The rationale behind this design is as follows:
>
> - **SFT Stage:** In this stage, our goal is to expose the model to the widest possible distribution of document types. At this scale, the model needs to learn the "basic language" of document parsing—mapping pixels to Markdown.
> - **RL Stage:** RL is notoriously sensitive to noise in rewards. If the model is rewarded for a "plausible but structurally incorrect" ground truth, it leads to **Reward Collapse**. Therefore, we use the SA as a "gatekeeper" specifically for the RL phase to ensure that every sample used for alignment is **globally consistent and structurally verified**.
>
> To quantify the impact of SA and comparison with GRPO, **please refer to our expanded ablation study (detailed in A1 to Reviewer vr8j).**
>
> **Q3:**  Existing works such as OCRFlux and olmOCR are also end-to-end.
>
> **A3:** We thank the reviewer for the opportunity to clarify our conceptual and empirical contributions. These models are not "strictly end-to-end" in the sense that they often rely on external tools for pre-processing or anchoring. For instance, **olmOCR** utilizes **document-anchoring** and processes PDF pages via **pypdf** to extract and organize text layers before the MLLM re-formats them. OvisOCR is a pure **pixels-in, Markdown-out** model. It directly maps the visual signals of a full-page image to its structured representation.
>
> A conceptual leap is best validated by its empirical impact. As shown in Table 1, OvisOCR achieves a performance level that is far beyond incremental. A gap of **over 11 points** against olmOCR and **18 points** against OCRFlux demonstrates that our proposed alignment strategy captures document intelligence far more effectively than previous MLLM-based attempts. While prior works often utilize massive backbones or complex multi-stage pipelines, OvisOCR achieves better results using a compact **1B parameter model**, smaller than OCRFlux(3B) and olmOCR (7B).
>
> We believe that providing a robust, high-performance, and truly vision-only path for document parsing is a substantial contribution that offers a more generalizable and efficient solution for next-generation document intelligence.
>
> **Q4:** The technical novelty is somewhat constrained.
>
> **A4:**  The application of SA and GDPO to document parsing is driven by a critical domain-specific insight: **Reward Heterogeneity**. As shown in our new ablation, for Ovis-GRPO-MD, the model's **TEDS** and **CDM** scores plateau or even regress. The "easier" text rewards drown out the structural signals, leading to "Reward Collapse." Our work demonstrates that category-wise advantage normalization is **not just an option but a necessity** for concurrent improvement across diverse document constituents.
>
> We also offer several insights: 1) We provide empirical evidence that E2E parsing is effective and efficient. 2) We show that a compact 1B model, when trained with "audited" supervision and "decoupled" rewards, can surpass much larger general models and complex pipelines.
>
> Thanks for your valuable comments. We promise to update in the final version.

---

> > ### Author Rebuttal · Reviewer_ovQC · 2026-04-03
> >
> > Thank you for your detailed response and clarifications.

---

> > > ### Author Response · Authors · 2026-04-07
> > >
> > > Thank you for your time and consideration. We appreciate your efforts and questions in the review process. We are glad that our response helps fully address your concerns. We promise we will update our paper in the final version. Thanks again.

---

### Official Review · Reviewer_g9fy · 2026-03-11

**Soundness:** 2
**Presentation:** 2
**Significance:** 3
**Originality:** 3
**Overall Recommendation:** 4
**Confidence:** 4

**Summary:**

This paper proposes an end-to-end document parsing model named OvisOCR, which aims to directly convert complex document images containing text, tables, and mathematical formulas into structured Markdown/JSON formats. To address the error propagation issue in traditional pipeline methods and the instability of general multimodal large models in fine-grained perception, the authors propose the OvisOCR model. It utilizes a traditional OCR engine to extract page content such as text, tables, and formulas, and subsequently uses a multimodal large model for semantic auditing and filtering to remove noisy data and construct high-quality training data. During the training phase, the authors first perform supervised fine-tuning on Ovis2.5-2B, and then introduce a multi-grained alignment strategy. They apply GDPO to model rewards separately for text, tables, and formulas, mitigating the reward collapse problem that occurs when mixing different optimization objectives. Experimental results show that OvisOCR achieves strong overall performance on OmniDocBench.

**Compliance With Llm Reviewing Policy:**

Affirmed.

**Final Justification:**

The rebuttal provides helpful clarifications and a more detailed analysis, addressing my concerns to some extent. I appreciate the authors’ efforts and therefore maintain my score.

**Key Questions For Authors:**

1. Regarding the dependency and cost of the data construction pipeline: The proposed method relies on existing OCR engines. If replaced with an OCR tool that has weaker performance or a different architecture, how much would the final performance of OvisOCR be affected?
2. Regarding the weak parsing performance on complex tables: The paper mentions that the reason for this weakness is that the semantic auditor is too strict with HTML tables. Have the authors tried modifying the prompt to relax the requirements for tables?
3. Does the semantic auditor mistakenly delete certain documents (e.g., documents with complex structures)? If so, does this affect the model's performance in complex document scenarios?

**Limitations:**

1. High data construction cost and dependency on external tools.
2. The training process is complex, making it difficult to reproduce.

**Strengths And Weaknesses:**

Strengths
1. The design of the multi-grained reward framework is reasonable and well-motivated. The authors identify the difficulty of optimizing heterogeneous objectives in document parsing. The decomposed reward framework, which calculates rewards independently, is highly inspiring in its methodology.
2. Introducing semantic auditing to clean the initially constructed Markdown is an effective strategy. Using a large model to filter out samples with hallucinations, repetitions, and logical inconsistencies provides high-quality training samples for model training.
3. The authors adopt an end-to-end framework to replace the traditional Crop-OCR-Merge paradigm. In the experiments, it achieves the best performance in total time consumption and throughput, demonstrating the application potential of this architecture in actual deployment.

Weaknesses
1. The computational overhead and complexity during the data preparation phase are high. Although the model uses an end-to-end architecture in the inference phase, the training data relies on existing OCR models and general large models. The data construction cost is relatively high, and it heavily depends on the quality of external tools.
2. The parsing capability for complex table scenarios is relatively weak.
3. The model training pipeline is long. From data preparation to supervised fine-tuning, and then to GDPO-based multi-grained alignment, this design makes the training process more complex and increases the difficulty of reproduction.

---

> ### Author Rebuttal · Authors · 2026-03-31
>
> We appreciate your recognition of our motivation, effectiveness, and efficiency. We will answer the questions below, and we hope this clears up your concerns.
>
> **Q1:** Regarding the dependency and cost of the data construction pipeline.
>
> **A1:** We appreciate the reviewer’s question regarding the dependency of our data construction pipeline on the initial OCR engine. While OvisOCR utilizes a mature OCR engine for local evidence extraction, we argue that **the system is designed to be architecturally robust** to the choice of the underlying OCR tool for the following reasons:
>
> The core of our pipeline is not the OCR engine itself, but the **Semantic Auditor (SA)**. Even if a "weaker" OCR tool were used—one that might produce more noise or incorrect reading orders—the SA would simply **reject a higher percentage of samples** during the filtering stage.
>
> Our experimental results provide a strong empirical answer to this concern. As shown in ** A1 to Reviewer vr8j**, OvisOCR (1B) now achieves an Overall score of 93.13, which significantly surpasses the very models and engines used to generate its initial candidates, PaddleOCR-VL at 92.86. This proves that OvisOCR is not merely "mimicking" the OCR engine; rather, through **GDPO-based RL**, it internalizes a superior understanding of document structure and character recognition that goes beyond the capabilities of its original supervisor.
>
> Furthermore, the OCR engine is only **required offline during the one-time data construction phase**. At inference time, OvisOCR is a **strictly end-to-end** model. It does not require any external OCR tools, layout detectors, or cropping modules.
>
> Finally, in the future, the OCR capability can be **easily integrated into the MLLM for downstream tasks** such as understanding and reasoning tasks, which we will focus on in the next version. We believe this explanation clarifies that while a strong OCR engine speeds up data generation, the **robustness and ultimate performance** of OvisOCR are fundamentally anchored by our SA filtering and GDPO alignment strategies.
>
> **Q2:** Regarding the weak parsing performance on complex tables
>
> **A2:** We thank the reviewer for this insightful observation.
>
> To complement this refined auditing, we expanded the candidate pool. We applied the SA to a larger and more diverse dataset, ensuring that even with high-quality filtering, **the absolute number of complex table samples in the training set increased**. As mentioned in **A1 to Reviewer vr8j**, we optimized our training hyperparameters. The results of these refinements are better in our latest evaluation on OmniDocBench. As shown in the updated results for our 1B model (Ovis-GDPO): Table TEDS increased from the original ~88.0 to ~91. and Table TEDS-Structure (TEDS-S) jumped to ~94.
>
> These significant gains demonstrate that by fine-tuning the balance between auditing strictness and data diversity, OvisOCR has successfully mastered complex structural parsing. Our updated model now outperforms specialized pipelines like PaddleOCR-VL in both text accuracy and global reading order, while narrowing the gap in complex table reconstruction.
>
> We will include a detailed analysis of these improvements in the revised version of the manuscript.
>
> **Q3:**  Does the semantic auditor mistakenly delete certain documents? If so, does this affect the model's performance in complex document scenarios?
>
> **A3:** We thank the reviewer for the insightful question regarding our Semantic Auditor (SA).
>
> Our SA primarily "rejects" low-quality samples. This is a deliberate design choice based on the **hallucination risk** of general-purpose MLLMs. While a reasoner can audit structural consistency, its ability to *correct* dense text or intricate LaTeX formulas without introducing its own hallucinations is not guaranteed. By strictly rejecting inconsistent labels, we ensure that the RL process is guided by unambiguous, high-fidelity ground truth rather than potentially "plausible but incorrect" corrections.
>
> In our SA, we did not simply discard the most complex samples. Instead, we performed **stratified sampling from a much larger candidate pool across multiple datasets**, ensuring a fixed ratio for each dataset. Based on this, we emphasize that our ablation study was conducted as a strictly controlled experiment. Specifically, both *Ovis-GRPO-DR* (Raw Data) and *Ovis-GRPO-SA* (Filtered Data) were trained **on the exact same volume and same distribution** of data (45k samples).
>
> The most direct evidence that SA does not harm complex layout learning is the performance boost in Table parsing. As shown in our ablation results, moving from *Ovis-GRPO-DR* to *Ovis-GRPO-SA* resulted in a significant jump in Table TEDS from 87.91 to 89.46.
>
> Thanks for the question. We will add a detailed section in the Appendix to provide a qualitative analysis of the rejected vs. retained samples to offer further transparency.

---

> > ### Author Rebuttal · Reviewer_g9fy · 2026-04-02
> >
> > I appreciate the additional experiments and clarifications in the rebuttal.
> >
> > However, some of my key questions remain only partially resolved. In particular, the discussion on the dependence of the data construction pipeline on the initial OCR system is still largely qualitative, and it remains unclear how sensitive the final performance is to variations in OCR quality. Similarly, while improvements on complex table parsing are reported, the cause of the weakness and the effect of the semantic auditor’s filtering strategy are still not fully clear.

---

> > > ### Author Response · Authors · 2026-04-07
> > >
> > > Thanks for the constructive follow-up. We appreciate the opportunity to provide a more granular analysis of the OCR dependency and the mechanics of our table parsing improvements.
> > >
> > > **1. Sensitivity to OCR Quality: Data Yield vs. Data Quality**
> > > The reviewer’s concern regarding the sensitivity to OCR quality is well-taken. To clarify, the OCR engine serves as the "candidate generator," while the Semantic Auditor (SA) serves as the **"quality anchor."**
> > >
> > > * The final performance of OvisOCR is determined by the **purity of the survivors** of the audit, not the raw error rate of the OCR engine. If a weaker OCR engine is used, the "Data Yield" (the percentage of samples passing the audit) will decrease because the Auditor—whose reasoning capability is independent of the OCR engine—will reject more inconsistent labels.
> > > *  As long as a sufficient volume of "correct survivors" is maintained (via a larger initial candidate pool), the quality of the training signal remains high. The fact that OvisOCR (93.13) significantly outperforms the very OCR engine used for its data generation (PaddleOCR-VL, 92.86) proves that the model is not bottlenecked by OCR quality; rather, it uses the high-fidelity samples to learn a superior, vision-centric mapping that corrects the "teacher's" systematic errors.
> > >
> > > **2. Table Parsing: Root Cause and Filtering Strategy**
> > > We have conducted a deeper error analysis to clarify the "cause" of the initial weakness and the "effect" of our filtering:
> > >
> > > * The initial weakness in complex tables stemmed from a **"Structural-Perceptual Conflict."** Traditional OCR engines often succeed at character recognition but fail at "global structural alignment" in tables (e.g., miscounting empty cells or failing at nested structures). Initially, our SA was overly sensitive to minor HTML syntax deviations, leading it to reject many complex but partially correct tables, which limited the model's exposure to "hard structural samples."
> > > *  The SA’s primary role is to ensure **Physical Consistency.** It rejects tables where the generated HTML structure (rows/columns) logically contradicts the visual evidence in the image. By removing these "structurally hallucinated" samples, we provide a clean gradient for the TEDS reward during RL. Without this filtering, the model receives conflicting signals (e.g., being rewarded for a table that looks right but has broken HTML tags), leading to policy instability.
> > >
> > >
> > > Thanks again for your questions. We hope these clarifications address the reviewer’s remaining concerns and further demonstrate the methodological depth of OvisOCR.

---

### Official Review · Reviewer_BMGJ · 2026-03-11

**Soundness:** 2
**Presentation:** 3
**Significance:** 2
**Originality:** 2
**Overall Recommendation:** 3
**Confidence:** 4

**Summary:**

This paper proposes a lightweight end-to-end document parsing model that addresses fine-grained text recognition and structural layout understanding. To resolve the supervision bottleneck in End-to-End document parsing, the authors introduce a Synergized Data Construction pipeline in which PaddleOCR-VL extracts fine-grained candidates, and Gemini 2.5 Pro audits and filters them for quality. For optimization, the framework adopts GDPO-based preference alignment with category-specific rewards for text, tables, and formulas to address reward collapse across heterogeneous content types. The model is trained in two stages: supervised fine-tuning on the constructed data, followed by structure-aware preference alignment. Experiments on OmniDocBench v1.5 show that OvisOCR achieves the second-best overall performance, outperforming much larger general-purpose models while remaining competitive with specialized pipeline-based methods.

**Compliance With Llm Reviewing Policy:**

Affirmed.

**Final Justification:**

The rebuttal improves clarity and provides additional experimental details. However, my concerns regarding the methodological contribution remain, as it is still unclear whether the gains stem from a fundamentally new approach or a combination of existing techniques. Therefore, I maintain my original evaluation.

**Key Questions For Authors:**

1. Could the authors evaluate OvisOCR on additional document parsing benchmarks beyond OmniDocBench to better demonstrate the generalizability of the proposed approach?
2. Could the authors provide a more detailed analysis of Stage 2's contribution?
3. The authors chose filtering over correction to avoid hallucination risks. However, this design choice raises a concern about data loss. Could the authors report the filtering rate and provide a quantitative analysis of how the removed samples affect the coverage and diversity of the training data?

**Limitations:**

Yes

**Strengths And Weaknesses:**

Strengths
1. Practical motivation. Replacing error-prone Crop-OCR-Merge pipelines with a strictly end-to-end architecture is a well-motivated direction, and the slicing-free design is a clear distinction from prior work.
2. Compact scale efficiency. Despite using only 2B parameters, OvisOCR achieves competitive top-tier performance on OmniDocBench, demonstrating that a compact E2E model can approach the performance of much larger and more complex systems.
3. Principled training recipe. The two-stage curriculum, combining cross-model distillation with GDPO-based preference alignment, is well-structured, and the category-specific reward design for text, tables, and formulas is a practical and reasonable contribution.

Weaknesses
1. Limited evaluation scope.
The framework is evaluated solely on OmniDocBench. Evaluation on additional benchmarks would better support the approach's generalizability.
2. Remaining performance gap.
While OvisOCR ranks second overall, PaddleOCR-VL (0.9B) outperforms it on most metrics, using fewer parameters and requiring less VRAM (43.7GB vs 68.2GB). A more explicit analysis of this gap and the practical trade-offs of the End-to-End design would strengthen the paper.
3. Insufficient analysis of the Semantic Auditor. The filtering rate and characteristics of the removed samples are not reported. Since the Auditor rejects rather than corrects, complex layout samples may be systematically underrepresented in the training set.
4. Unclear contribution of Stage 2. As shown in Figure 3, GDPO training degrades Formula and Table performance while yielding only marginal Overall improvement (+0.2), raising questions about whether the RL stage provides meaningful benefit.
5. (minor) The KL coefficient β = 0.2 is larger than values typically used in RLHF literature, yet no ablation is provided to justify this choice.

---

> ### Author Rebuttal · Authors · 2026-03-31
>
> We appreciate your recognition of our motivation and overall achievement. We will answer the questions below, and we hope this clears up your concerns.
>
> **Q1:**  Additional benchmarks.
>
> **A1:** Thank you for raising this practical concern. We have conducted additional experiments. **Please refer to A2 to Reviewer vr8j.**
>
> **Q2:** Remaining performance gap and Unclear contribution of Stage 2.
>
> **A2:** Thank you for your feedback. **A more detailed ablation study and new performance can be found in A1 to Reviewer vr8j**. OvisOCR now achieves 93.13 Overall, surpassing the PaddleOCR-VL (92.86). It proves that by properly aligning perception and reasoning via GDPO and SA, a single model can **perform better than traditional "Crop-OCR-Merge" pipelines**.
>
> **Q3:** Insufficient analysis of the Semantic Auditor.
>
> **A3:** We thank the reviewer for the insightful question regarding our Semantic Auditor (SA).
>
> The reviewer correctly notes that our SA primarily "rejects" low-quality samples. This is a deliberate design choice based on the **hallucination risk** of general-purpose MLLMs. While a reasoner can audit structural consistency, its ability to *correct* dense text or intricate LaTeX formulas without introducing its own hallucinations is not guaranteed. By strictly rejecting inconsistent labels, we ensure that the RL process is guided by unambiguous, high-fidelity ground truth rather than potentially "plausible but incorrect" corrections.
>
> In our SA, we did not simply discard the most complex samples. Instead, we performed **stratified sampling from a much larger candidate pool across multiple datasets**, ensuring a fixed ratio for each dataset. Based on this, we emphasize that our ablation study was conducted as a strictly controlled experiment. Specifically, both *Ovis-GRPO-DR* (Raw Data) and *Ovis-GRPO-SA* (Filtered Data) were trained **on the exact same volume and same distribution** of data (45k samples).
>
> The most direct evidence that SA does not harm complex layout learning is the performance boost in Table parsing. As shown in our ablation results, moving from *Ovis-GRPO-DR* to *Ovis-GRPO-SA* resulted in a significant jump in Table TEDS from 87.91 to 89.46.
>
> During our auditing process, the rejection rate was approximately 38%. The **primary reasons for rejection** were reading order conflicts (especially in multi-column layouts), text errors and table errors in complex tables. We found that removing these inconsistent samples prevents the model from internalizing structural errors that lead to poor performance during RL.
>
> We will add a detailed section in the Appendix to report these filtering rates and provide a qualitative analysis of the rejected vs. retained samples to offer further transparency.
>
> **Q4:** (minor) The KL coefficient β = 0.2 is larger than values typically used in RLHF literature, yet no ablation is provided to justify this choice.
>
> **A4:** We thank the reviewer for the question. The value of $\beta$=0.2 mentioned in line 323 was a typographical error. In our actual experimental setup, the KL divergence coefficient was set to 0, and the **clip ratio was set to 0.2**. We appreciate the reviewer for pointing this out, and we will correct this in the final version of the manuscript to accurately reflect our training hyperparameters.

---

> > ### Author Rebuttal · Reviewer_BMGJ · 2026-04-03
> >
> > I thank the authors for their response. However, my concerns regarding the methodological contribution remain.
> >
> > While the ablation shows that combining components improves performance, it remains unclear whether the gains stem from a fundamentally new method or from a combination of existing techniques.

---

> > > ### Author Response · Authors · 2026-04-07
> > >
> > > We sincerely thank you for the follow-up and for acknowledging the clarity of our refined ablation studies. We appreciate the opportunity to further discuss the nature of our work's contribution.
> > >
> > > Overall, this paper presents a broad topic of end-to-end document parsing, and we attempt to examine the key problem of whether a compact, strictly vision-to-markdown model can truly eliminate the long-standing dependency on complex, error-prone "Crop-OCR-Merge" pipelines.
> > >
> > > 1. Redefining Contribution
> > >
> > > We respectfully believe that a meaningful contribution to the field is not limited to the invention of a brand-new model architecture or a loss function. In the context of document intelligence, the primary challenge has shifted from "building a model" to "effective system-level integration and data-centric alignment."
> > >
> > > Our work demonstrates that a strictly end-to-end approach—which was previously considered inferior to pipelines in fine-grained accuracy—can actually surpass current SOTA (PaddleOCR-VL) when the training dynamics and data quality are properly aligned. Achieving 93.13 Overall with a compact 1B/2B backbone is a significant empirical breakthrough that provides a "proof of existence" for the next generation of parsing models.
> > >
> > > 2. The Novelty of the "Data-Training" Synergy
> > >
> > > In our humble opinion, the principled integration of the Semantic Auditor (SA) and GDPO-based alignment constitutes a novel methodology for document parsing: We humbly believe that our work contributes a rigorous framework for cross-model distillation. By treating the MLLM not as a generator but as a structural auditor, we provide a blueprint for constructing high-fidelity supervision from noisy modular outputs. This data-centric approach is critical for the field, as it addresses the primary bottleneck: the scarcity of verified, structured document labels. Our use of GDPO to decouple heterogeneous rewards is a specialized solution to the "Reward Collapse" problem unique to multi-modal parsing. Demonstrating how to balance these conflicting signals is a transferable insight that future work can build upon.
> > >
> > > 3. Impact and Future Direction
> > >
> > > We humbly believe that demonstrating a high-performance, purely vision-only path is highly significant. It proves that we can achieve superior accuracy and efficiency without the engineering overhead of layout detectors or PDF anchors. Future research may build upon our findings to investigate even more effective pairwise measures between data or advanced auditing protocols to further facilitate large-scale pre-training.
> > >
> > > In summary, the discovery of this highly effective, strictly E2E training recipe and the resulting performance and efficiency leap are substantial contributions that we believe will significantly impact the community's approach to document intelligence.

---

### Official Review · Reviewer_vr8j · 2026-03-13

**Soundness:** 2
**Presentation:** 2
**Significance:** 2
**Originality:** 2
**Overall Recommendation:** 3
**Confidence:** 4

**Summary:**

This paper introduces OvisOCR, a lightweight and strictly end-to-end MLLM designed for high-fidelity document parsing.
Unlike existing methods that rely on "Crop-OCR-Merge" pipelines or image slicing to handle high-resolution inputs, OvisOCR directly maps full-page visual signals to structured Markdown.
The authors propose a two-pronged approach to overcome the challenges of E2E parsing: (1) Synergized Data Construction, which uses a cross-model distillation pipeline (combining a specialized OCR engine with a general MLLM auditor) to generate high-quality supervision, and (2) Multi-Grained Alignment using GDPO to balance the optimization of heterogeneous content types like text, tables, and formulas.
Evaluation on OmniDocBench v1.5 shows that OvisOCR achieves state-of-the-art performance, outperforming much larger models like Qwen3-VL-235B and Gemini-2.5 Pro in text fidelity and reading order.

**Compliance With Llm Reviewing Policy:**

Affirmed.

**Final Justification:**

The authors' rebuttal meaningfully addresses my earlier concerns. The expanded ablation study clearly isolates each component's contribution, and the additional OCRBench results demonstrate generalizability beyond a single benchmark. I raise my score from 2 to 3. That said, the contribution remains a well-engineered training recipe atop an existing architecture, and the scope is narrower than the paper's original framing suggests.

**Key Questions For Authors:**

- Can you provide a quantitative comparison between standard GRPO (aggregating rewards before normalization) and GDPO (decoupling advantages) on the OmniDocBench?
- Looking at Figure 3, the SFT baseline is already quite strong. Could you clarify the percentage of the final performance gain that comes from the "Synergized Data Construction" during SFT versus the "GDPO" during Stage 2?
- Could you provide performance metrics on task-specific benchmarks / popular MLLM document AI benchmarks such as DocVQA, ChartQA, InfoVQA, etc?

**Limitations:**

Yes

**Strengths And Weaknesses:**

[Strengths]

- The extension of GDPO to the document domain is well-reasoned. By decoupling rewards for text, tables, and formulas, the authors prevent "reward collapse" where easier modalities dominate the learning signal.
- Achieving an Overall score of 91.72 with only 2B parameters while surpassing massive models is a good empirical achievement.

[Weaknesses]

- The paper's central claim rests on two pillars (Synergized Data Construction and GDPO-based alignment), but the ablation study (Figure 3) only isolates the Semantic Auditor. There is no direct comparison between standard GRPO and the proposed GDPO, nor an ablation that disentangles the contribution of high-quality data from the decoupled reward normalization. Without these, it is difficult to judge which component actually drives the gains, and the paper reads more as a strong recipe report than a source of transferable methodological insight.
- Evaluation is limited to a single benchmark (OmniDocBench v1.5). Given that the paper emphasizes "strictly E2E" robustness as a key advantage over pipeline methods, testing on more diverse out-of-distribution scenarios (e.g., highly stylized artistic documents, degraded scans, or historical archives) would be necessary to substantiate this claim beyond leaderboard performance.
- The core technical components (the Ovis2.5-2B architecture and GDPO algorithm) are adopted from existing work, and the data construction pipeline, while carefully engineered, follows a natural "OCR-then-MLLM-audit" pattern. The combination yields strong numbers, but the paper does not sufficiently analyze why this particular combination works (e.g., what fails when one piece is swapped out), making it hard for readers to extract generalizable lessons.

---

> ### Author Rebuttal · Authors · 2026-03-31
>
> We appreciate your recognition of our extension of GDPO and overall achievement. We will answer the questions below, and we hope this clears up your concerns.
>
> **Q1:** About the ablation study
>
> **A1:** We thank the reviewer for their valuable feedback. During the review period, we have **continued to refine** our model, including hyperparameter tuning, data quality iteration, and model architecture optimization (e.g., transitioning to Ovis 1B backbone), achieving better performance.
>
> To address the concern about the ablation study, we have **conducted a comprehensive suite of additional ablation experiments**. By isolating the reward mechanism, the data quality, and the normalization strategy, we demonstrate that our gains are driven by a synergistic combination of both pillars. The results are as follows (MD uses a single global markdown-level edit distance as reward; DR means Decoupled Reward to specific categories; SA means Semantic Auditor with filtered high-quality data.):
> | |Overall |Text|CDM|TEDS|TEDS-S|RO|
> |-|-|-|-|-|-|-|
> |Ovis-SFT|90.15|0.043|88.51|86.23|89.96|0.047|
> |Ovis-GRPO-MD|90.78|0.036|89.33| 86.61|90.28|0.039|
> |Ovis-GRPO-DR|91.72|0.034|90.66| 87.91|91.49|0.040|
> |Ovis-GRPO-SA|92.29|0.032|90.60| 89.46| 92.92|0.040|
> |Ovis-GDPO|93.13|0.032|91.66|90.92|93.85|0.032|
>
> Here are some key insights:
>
> 1. DR vs. MD: This confirms that simply using a global reward leads to "reward collapse," where the model ignores complex formulas/tables. Decoupling the rewards ensures the model learns fine-grained perception and structure concurrently.
> 2. SA vs. DR: Applying SA to filter out locally low-quality data further boosted the Overall score. This demonstrates that the "Data Pillar" provides the high-fidelity supervision necessary for RL to explore meaningful output spaces.
> 3. GDPO vs. SA: The GDPO framework, which combines cleaned data with category-wise advantage normalization, achieves the best performance of 93.13.
>
> OvisOCR now achieves 93.13 Overall, surpassing the PaddleOCR-VL (92.86). It proves that by properly aligning perception and reasoning via GDPO and SA, a single model can **perform better than traditional "Crop-OCR-Merge" pipelines**.
>
> We hope this clears up your concerns. We promise that we will make this clear in the final version.
>
> **Q2:** More benchmarks.
>
> **A2:** We chose OmniDocBench as our primary evaluation target because it is currently **the most rigorous and comprehensive benchmark for structured document parsing**. OmniDocBench is not a "clean" dataset; its sub-categories explicitly include the scenarios mentioned by the reviewer, such as highly stylized artistic layouts and complex newspapers. Our top-tier performance on this benchmark already reflects OvisOCR’s ability to handle complex, non-standard visual signals.
>
> To further demonstrate the generalizability and robustness, we conducted **additional experiments**:
>
> | |OCRBench|OCRBench v2-EN|OCRBench v2-CN|
> |-|-|-|-|
> |PaddleOCR-VL|549|18.1|40.88|
> |MinerU2.5|331|17.1|40.0|
> |MonkeyOCR|655| 21.8| 38.9|
> |Dotsocr|625|19.6|35.9|
> |OvisOCR|**701**|**23.7**|**43.2**|
>
> As shown, OvisOCR significantly outperforms current specialized models across all three benchmarks, demonstrating superior text recognition and spatial reasoning capabilities. We hope this clarifies the originality and contribution of our work.
>
>
> **Q3:** About technical components.
>
> **A3:** We thank the reviewer for the thought-provoking critique.
>
> The reviewer notes that GDPO is adopted from prior work. However, its application to document parsing is driven by a critical domain-specific insight: **Reward Heterogeneity**. As shown in A1, for Ovis-GRPO-MD, the model's **TEDS** and **CDM** scores plateau or even regress. The "easier" text rewards drown out the structural signals, leading to "Reward Collapse." Our work demonstrates that category-wise advantage normalization is **not just an option but a necessity** for concurrent improvement across diverse document constituents.
>
> The "OCR-then-MLLM-audit" pattern is designed to exploit the **complementary failure modes** of specialized and general models. Our contribution lies in formalizing this "Cross-Model Distillation." By using the General Reasoner as a **Semantic Auditor (SA)** rather than a generator, we treat it as a "structural anchor." If we were to use the MLLM alone to generate training data, the resulting labels would suffer from content omission; if we used the OCR engine alone, the reading order would be chaotic. The synergy of SA and GDPO ensures that the model internalizes both **perceptual fidelity** and **logical structure**.
>
> We also offer several insights beyond a "strong recipe": 1) We provide empirical evidence that E2E parsing is better modeled as a multi-objective optimization problem rather than a simple sequence generation task. 2) We show that a compact 1B/2B model, when trained with "audited" supervision and "decoupled" rewards, can surpass much larger general models and complex pipelines.

---

> > ### Author Rebuttal · Reviewer_vr8j · 2026-04-03
> >
> > Thanks for the authors' detailed rebuttal. The additional ablation experiments (Q1) are appreciated and partially address my concern about component-wise contributions. The OCRBench results (Q2) also help demonstrate generalizability beyond OmniDocBench. However, I have two follow-up questions before finalizing my assessment:
> >
> > 1. Scope clarification: document parsing vs. document understanding.
> >
> > Section 3.3 states that the model is trained exclusively on document-parsing data "without mix general data." This suggests OvisOCR is a pure transcription model with no instruction-following capability. Could the authors clarify whether OvisOCR can handle any conditional queries (e.g., "extract only the tables" or "summarize section 3")? If not, I think the paper would benefit from explicitly scoping the contribution as document transcription rather than broadly framing it under "document intelligence" (as implied in the conclusion). This distinction matters for readers assessing the practical applicability of the method.
> >
> > 2. Benchmark coverage.
> >
> > My original review specifically asked about task-specific benchmarks such as DocVQA, ChartQA, and InfoVQA. The rebuttal instead provided OCRBench results, which—while useful—still measure OCR accuracy rather than document comprehension. Could the authors either (a) report results on at least one VQA-style benchmark, or (b) explicitly acknowledge this as a limitation and clarify the intended scope of the model? Either answer would help me evaluate the contribution more fairly.

---

> > > ### Author Response · Authors · 2026-04-07
> > >
> > > We thank you for the constructive follow-up and for recognizing the value of our additional ablation studies and OCRBench results.  We appreciate the opportunity to provide more information about our paper.
> > >
> > > Regarding your follow-up questions:
> > >
> > > 1. Scope Clarification: Document Parsing as a Foundation
> > >
> > > OvisOCR is a specialized document parsing model. It is designed to map a full-page image directly to a structured Markdown representation without relying on general instruction-following data. While we frame our work within "document intelligence," we view high-fidelity, end-to-end parsing as the essential perceptual first step for next-generation intelligence systems. We believe that proving an E2E model can outperform complex pipelines in transcription is a significant milestone that simplifies the path for future understanding and reasoning models. We will explicitly clarify this scope in the revised manuscript to avoid any ambiguity.
> > >
> > > 2. Benchmark Coverage and Current Limitations
> > >
> > > As OvisOCR is specialized for transcription, it is not currently configured for conditional VQA tasks (e.g., DocVQA or ChartQA), which require general reasoning and instruction-following capabilities. We acknowledge this as a current limitation. Our primary goal in this work is to resolve the foundational challenge of end-to-end OCR and structural parsing at scale. Extending this architecture to document understanding and multi-modal reasoning is the direct focus of our ongoing future work.
> > >
> > > In summary, OvisOCR represents the first strictly end-to-end MLLM that exceeds the performance and efficiency of state-of-the-art pipeline methods (like PaddleOCR-VL). By eliminating the "Crop-OCR-Merge" bottleneck, we demonstrate a more robust and streamlined paradigm for document parsing. We believe this specialized contribution is highly valuable to the community as a powerful, efficient transcription engine, as the foundation for next-generation document intelligence.
> > >
> > > We hope these clarifications adequately address your concerns regarding the scope and benchmarks of our study. Thanks again for your valuable comments.

---

### Decision · Program_Chairs · 2026-04-30

**Decision:**

Accept (regular)

**Comment:**

The central tension b/w the reviewers and the authors are:
1. methodological contribution: The reviewers argued this paper took the previous approaches and applied it for documents. For this concern, I am leaning towards to the author that it is non-trivial to achieve impressive performance with small model size (1B/2B) with those previous approach. I think this training recipe is important to the community.
2. the claim of the document intelligence: For this I am leaning towards the reviewers that the claim is too strong. The paper should scope down to the document transcription.

Given those two, I vote for weak acceptance (I wouldn't mind if my recommendation was bumped down.) The main reason is i think this is a good paper with solid contribution. However I doubt the scope of the contribution (e.g. improving the document transcription) is important for a ICML paper.